# Variations in the physicochemical and optical properties of natural aerosols in Puerto Rico - Implications for climate

Héctor Rivera<sup>1</sup>, John A. Ogren<sup>2,3</sup>, Elisabeth Andrews<sup>3</sup>, Olga L. Mayol-Bracero<sup>4</sup>

<sup>1</sup>Department of Physics, University of Puerto Rico - Rio Piedras, San Juan, Puerto Rico <sup>2</sup>Earth Systems Research Laboratory, National Oceanic and Atmospheric Administration, Boulder, Colorado <sup>3</sup>Cooperative Institute for Research in Environmental Sciences, University of Colorado, Boulder, Colorado <sup>4</sup>Department of Environmental Sciences, University of Puerto Rico - Rio Piedras, San Juan, Puerto Rico

Correspondence to: Olga L. Mayol-Bracero: (omayol@ites.upr.edu)

- Abstract. Atmospheric aerosols stay a major cause of uncertainty in climate prediction. Hundreds of teragrams and the absorbing properties of aerosols such as African dust and volcanic ash affect radiative balance changing atmospheric temperature and thus, climate. Since 2005, we began to check the physicochemical and optical properties of aerosols at the Cape San Juan Atmospheric Observatory, Puerto Rico. Based on the Hybrid Single-Particle Lagrangian Integrated Trajectory backward trajectories and satellite imagery from the Volcanic Ash Advisory Center in Washington D.C.,
- Moderate Resolution Imaging Spectroradiometer, and Saharan air layer images, we grouped natural aerosols in three air masses: marine, African dust and volcanic ash. A sun-sky radiometer from the NASA's AErosol RObotic NETwork assessed total aerosol optical depth and its fine fraction. A 3-wavelength nephelometer and particle soot absorption photometer assessed the scattering and absorption coefficients. Two impactors segregated the submicron ( $D_p < 1 \mu m$ ) particles from the total ( $D_p < 10 \mu m$ ) enabling us to calculate the sub-micron scattering and absorption fractions. The measured variables
- served to calculate the single scattering albedo and radiative forcing efficiency. All variables but the single scattering albedo making up the aerosol climatology for Puerto Rico had different means as function of the air mass category at p<0.05. For the period 2005-2010, the largest means  $\pm$  95% confidence interval of the scattering coefficient (53  $\pm$  4 Mm<sup>-1</sup>), absorption coefficient (1.8  $\pm$  0.16 Mm<sup>-1</sup>), and optical depth (0.29  $\pm$  0.03), suggested African dust is the main contributor to the columnar and surface aerosol loading in summer. About two thirds (63%) of the absorption in African dust was due to the coarse mode
- and about one third due to the fine mode. In volcanic ash, fine aerosols contributed 60% of the absorption while coarse contributed 40%. Overall, the coarse and fine modes accounted for ~80% and 20% of the total scattering. The African dust load was 3.5 times the load of clean marine, 1.9 times greater than clean with higher sea salt content and 1.7 times greater than volcanic ash. African dust caused 50% more cooling that volcanic ash at the top of the atmosphere and 50% more heating than that of volcanic ash within the marine boundary layer.

## **1** Introduction

Atmospheric aerosols present high uncertainty in climate prediction (Boucher et al., 2013) because of their differences in amount and size, differing also in index of refraction that depends on the chemical composition and source. Different physicochemical properties of aerosols also result in diverse optical properties influencing climate and environment in many

- ways (Seinfeld and Pandis, 1998). Ogren (1995) pointed out that we need to evaluate the aerosols climate-forcing properties to know their spatial distribution, physical, optical, and cloud-nucleating properties, suitable radiative transfer models, and cloud physics. Anthropogenic as well as natural aerosol climate-forcing properties are important in this regard. Extensive research exists on the topic of anthropogenic aerosol climate forcing (Boucher et al., 2013); however, not so much has been done regarding natural aerosols, and even less is known about natural aerosols in the Caribbean, a region that plays a crucial
- role in global climate by serving as an atmospheric conduit between tropical and extra-tropical regions. The Caribbean is exposed to different type of natural aerosols such as those coming from marine sources, mineral dust from Africa and volcanic ash from the Soufriere Hills volcano in the island of Montserrat (e.g., Li-Jones and Prospero, 1998; Gioda et al., 2011; Prospero and Mayol-Bracero, 2013; Valle-Diaz et al., 2016; Wex et al., 2016). These three types of aerosols are the focus of this study.
- For marine aerosols, oceans produce the largest primary aerosol mass emissions (Warneck 1988) and are a key source of secondary atmospheric aerosols (O'Dowd and Smith, 1993; O'Dowd et al., 2004). The mass concentration and size distribution of marine aerosols depend on the wind speed (Woodcock, 1953; Lovett, 1978; Blanchard and Woodcock, 1980). Wind speed directly correlates with the sea-salt amount, but wind speed only explains part of the variance (Quinn and Coffman, 1999; Smirnov et al., 2003). Sea-salt is non-absorbing and comprises much of the marine boundary layer (MBL)
- aerosol mass, changing the radiative balance through scattering of visible light (Quinn et al., 1996; Winter and Chýlek, 1997). Also, sub-micron non-sea salt (nss) sulfate from biological activity scatters visible light efficiently also serving as cloud condensation nuclei (Charlson et al., 1991; Jacobson, 2001).

Turning to African dust, hundreds of teragrams reach the atmosphere every year (Huneuus et al., 2011), with large variations in emissions, in space and time (Prospero, 1999; Vinoj et al., 2004). These pace-time changing amounts result in poorly

- characterized African dust radiative-forcing properties (Liao and Seinfeld, 1998; Sokolik and Toon, 1999; Sokolik et al., 2001, which in turn reduces the accuracy of numerical models for predicting climate change (Houghton et al., 2001). In addition, African dust changes the radiative balance scattering and absorbing solar and terrestrial radiation. The dominant absorbing species in African dust are the iron oxides (Sokolik and Toon, 1999; Moosmüller et al., 2009).
- Such as in African dust, volcanic ash changes the radiative balance by scattering and absorbing solar radiation (Bohren and Huffman, 1999) because volcanic ash also holds iron oxides (Seinfeld and Pandis, 1998; Kokhanovsky, 2008). In addition, volcanic ash is hard, abrasive, and acidic (Krotkov et al., 1999; Housley et al., 2002) resulting in an aviation hazard. Evidence has shown engine failure in aircrafts flying through volcanic ash (Krotkov et al., 1999; USGS Fact Sheet, 2006).

Aerosols studies in the Caribbean goes back to the 1970's when Prospero et al. (1970) and Prospero and Carlson (1972) highlighted that synoptic outbreaks of Saharan dust occur from late spring to fall. These outbreaks extend from western Africa across the tropical Atlantic to the Caribbean. Other studies of transported African dust in the Caribbean, particularly in Puerto Rico, include those of Reid et al. (2002), Gioda et al. (2011), Fitzgerald et al. (2015), Spiegel et al. (2014); Denjean et al. (2016), Raga et al. (2016) and Valle-Diaz et al. (2016). From these studies, we highlight the Puerto Rico Dust Experiment (PRIDE) (Reid et al. 2003a, b), the only study that included the radiative, microphysical, and transport properties of African dust. Reid et al. (2002) reported that during the first half of PRIDE (June 2000), dust had the highest concentrations in the marine and convective boundary layers, with lower dust concentrations above the trade inversion

despite a strong Saharan Air Layer (SAL), a conceptual model by Karyampudi and Carlson (1988). PRIDE showed that

- coarse marine aerosols produced most of the scattering and optical depth in Puerto Rico. However, the African dust coarse mode generates most of the optical depth in spring and summer (Reid et al., 2003b). Also, with single-particle analyses E. A. Reid et al. (2003) reported that elemental iron composes ~2.5-3% of the total dust mass (assuming aluminium is 8% of the total mass). PRIDE excluded characterizing volcanic ash and characterizing the long-term variability of the climate-forcing properties of aerosols in Puerto Rico.
- In this article, we characterize the climate-forcing properties of natural aerosols in Puerto Rico. We analyse aerosol data collected in Puerto Rico from 2005-2010 to: 1) classify local aerosols by source (marine, African dust and volcanic ash), 2) characterize means and variabilities in climate-forcing properties of aerosols from these three natural sources and report the monthly climatology of aerosols in Puerto Rico, 3) test the hypothesis that "means and variability of aerosols from different sources differ significantly at p<0.05", and 4) determine if we can distinguish the kind of aerosol only knowing their mean</p>
- and variability in the climate-forcing properties.

# 2 Experimental

The sampling site was the Cape San Juan Atmospheric Observatory, at the natural reserve of Cabezas de San Juan, Puerto Rico (CPR), with coordinates (18°22.85'N, 65°37.07'W), managed by the Atmospheric Chemistry and Aerosols Research group at the University of Puerto Rico – Rio Piedras Campus, and supported by the Aerosol Group of the Global Monitoring

Division at the National Oceanic and Atmospheric Administration's Earth System Research Laboratory (NOAA/ESRL). CPR is a coastal site influenced by the trade winds most of the year where the absence of large land areas upwind reduces anthropogenic aerosols. African dust disturbs the marine environment of CPR from late spring to mid fall with stronger events in late spring and summer. Emissions from the Soufriere Hills volcano in Montserrat also disturb CPR if the low-level winds are southeast because Puerto Rico is about 400 km northwest of Montserrat.

#### 2.1 In-situ aerosol measurements

The aerosol measurements at CPR follow the NOAA Earth System Research Laboratory (Figure 1) network sampling protocols (<u>http://esrl.noaa.gov/gmd/aero/net/</u>), consistent with the guidelines of the World Meteorological Organization (WMO, 2016) aerosol measurements in the Global Atmosphere Watch. Delene and Ogren (2002) describe the aerosol

- monitoring. In Figure 1, a TSI model 3563 integrating nephelometer measures the aerosol scattering ( $\sigma_{sp}$ ) and backscattering ( $\sigma_{bsp}$ ) coefficients at 450, 550 and 700 nm. In addition, a Radiance Research Particle Soot Absorption Photometer (PSAP) measures the aerosol absorption coefficient ( $\sigma_{ap}$ ) at 467, 530, and 660 nm. The absorption at 530 nm was adjusted by log-log interpolation to 550 nm to yield  $\sigma_{ap}$  and  $\sigma_{sp}$  at the same wavelength (550 nm). Upstream of the nephelometer and PSAP a heater warms the aerosol sample to reduce the relative humidity to values around 40%. Two switched impactors segregate
- the aerosols in sub-micron ( $D_p < 1 \mu m$ ) and total ( $D_p < 10 \mu m$ ) fractions to compare the fine with the total aerosol contributions.

#### 2.1.1 Description of in-situ variables

Variables  $\sigma_{sp}$ ,  $\sigma_{ap}$  and extinction coefficient ( $\sigma_{ext}$ ) are extensive parameters (Ogren, 1995). Extensive parameters depend on aerosol amount and are additive. We report  $\sigma_{sp}$  and  $\sigma_{ap}$  in Mm<sup>-1</sup> (1 Mm<sup>-1</sup> = 10<sup>-6</sup> m<sup>-1</sup>).

- Calculating the sub-micron scattering (R<sub>sp</sub>) and absorption (R<sub>ap</sub>) fractions allowed us to test the contribution of the submicron mode to the total scattering or absorption. The R<sub>sp</sub> and R<sub>ap</sub> are intensive (i.e., independent of aerosol amount) and non-dimensional parameters associated with the scattering or absorbing particles size distributions (Ogren 1995). Intensive and extensive parameters are variables in chemical transport and radiative transfer models. We calculated R<sub>sp</sub> and R<sub>ap</sub> with Eqns. (1) and (2) from Delene and Ogren (2002).

$$R_{sp}(D_p) = \frac{\sigma_{sp}(D_p 

with  $\sigma_{sp}$  the scattering coefficient, B the scattering coefficient at a wavelength  $\lambda$  of one  $\mu$ m, and a the scattering Ångström exponent. We calculated the scattering Ångström exponent with Eqn. (4).

$$\mathbf{a} = -\frac{\log(\sigma_{\rm sp}^{550}/\sigma_{\rm sp}^{700})}{\log(\frac{550}{700})} \tag{4}$$

5

In Eqn. (4),  $\sigma_{sp}^{550}$  and  $\sigma_{sp}^{700}$  are the scattering coefficients at 550 and 700 nm. The scattering Ångström exponent qualitatively measures the sizes of scattering particles. The greater the Ångström exponent the smaller is the size of the scattering particles.

In addition, we calculated the single scattering albedo  $(\infty)$ , a non-dimensional intensive quantity, to estimate the contribution 10 of scattering to extinction with Eqn. (5).

$$\omega_0 = \frac{\sigma_{sp}}{\sigma_{ap} + \sigma_{sp}} \tag{5}$$

The  $\omega_0$  is part of the radiative forcing per unit of optical depth ( $\Delta RF/\delta_{ae}$ ) at the top of the atmosphere, called radiative forcing efficiency after Sheridan and Ogren (1999). The  $\Delta RF/\delta_{ae}$  depends on the aerosol size through the upscatter fraction  $\beta$ , the 15 scattering and absorption through  $\omega_0$ , and on seven geophysical quantities. We calculated the upscatter fraction with Eqn.  $\beta =$ 0.0817 + 1.8495b - 2.9682b<sup>2</sup>, where the backscatter fraction b was calculated with Eqn.  $b = \sigma_{bsp}/\sigma_{sp}$ . This parameterization, presented by Sheridan and Ogren (1999), omits the dependence of  $\beta$  with the zenith angle. The radiative forcing efficiency for daytime-average was calculated with Eqn. (6).

$$20 \quad \frac{\Delta RF}{\delta_{ae}} = -DS_0 T_{at}^2 (1 - A_c) \omega_0 \beta \{ (1 - R_s)^2 - \left( 2\frac{R_s}{\beta} \left[ \left( \frac{1}{\omega_0} \right) - 1 \right] \right) \}$$
(6)

Eqn. (6) assumes a constant geographical surface reflectance ( $R_s$ ) and atmospheric transmission ( $T_{at}$ ), with the values for the fractional day length (D) = 0.5, solar constant ( $S_o$ ) = 1370 W m<sup>2</sup>,  $T_{at}$  = 0.76, fractional cloud amount ( $A_c$ ) = 0.6, and  $R_s$  = 0.15 proposed by Haywood and Shine (1995). The aerosol measurements are at 550 nm (Delene and Ogren, 2002).

#### 25 2.2 Ground-based remote aerosol measurements

A CIMEL Electronique 318A spectral radiometer, part of NASA's AErosol RObotic NETwork (AERONET), assessed the total aerosol optical depth ( $\delta_{ae}$ ) and aerosol optical depth fine fraction ( $\delta_{ae}FF$ ). Holben et al. (1998) detail how to find  $\delta_{ae}$ . The total aerosol optical depth  $\delta_{ae}$  is the integral in the vertical of  $\sigma_{ext,\lambda}$ ,  $\delta_{ae}$  is a non-dimensional extensive parameter. AERONET finds the fine fraction with the Spectral Deconvolution Algorithm (SDA; O'Neill, 2001, 2003). Also, AERONET uses the

30 number size distributions to derive the volume size distribution through an inversion algorithm by Dubovik and King (2000).

We present volume size distributions to support other results. AERONET columnar measures are at 500 nm (green) to compare with the 550 nm (also green) at the surface.

#### 2.3 Models and satellite data

We used the Hybrid Single Particle Lagrangian Integrated Trajectory (HYSPLIT) Model (Draxler and Hess, 1998; Draxler
and Rolph, 2013) to estimate air mass trajectories from the sources to CPR. Moderate Resolution Imaging Spectroradiometer (MODIS) images served to sense possible African dust, volcanic emissions, or clean marine. SAL images proved useful to sense possible African dust over CPR. We used SAL images from <a href="http://tropic.ssec.wisc.edu/archive/">http://tropic.ssec.wisc.edu/archive/</a>. Images from the Volcanic Ash Advisory Center (VAAC) in Washington D.C. also served to sense volcanic ash if MODIS images were unavailable. VAAC obtains data from three Geostationary Operational Environmental Satellite (GOES) satellites (GOES-11, GOES-12, and GOES-13) covering from the central Pacific to the eastern Atlantic.

#### 2.4 Data processing and quality control

Nephelometer data were corrected for instrumental non-idealities, such as truncation error, after Anderson and Ogren (1998). The method corrects scattering measurements over integration angles of  $\sim 7^{\circ}-170^{\circ}$  and  $\sim 90^{\circ}-170^{\circ}$  to the full  $0^{\circ}-180^{\circ}$  and  $90^{\circ}-180^{\circ}$  ranges, based on the measured scattering Ångström exponent (å). The scattering Ångström exponent qualitatively

- 15 describes the scattering particles sizes (Anderson and Ogren, 1998). Concerning particles larger than the scattered light wavelength, the truncation error is greater than that of other systematic errors and can become twice larger. Uncertainties also arise because the nephelometer is calibrated with a gas (Carbon dioxide), which scatters in a Rayleigh regime, and aerosols scatter in the Mie regime. Scattering from gas molecules (Rayleigh scattering) is subtracted from the total scattering to find scattering by aerosol particles. Corrections to PSAP data were based on Bond et al. (1999), reporting a 20-30%
- 20 overall overestimation of absorption measured by the PSAP due to scattering aerosols on the filter before applying correction. Ogren (2010) extended the Bond et al. (1999) corrections to apply to measurements of a 3-wavelength PSAP. We manually edited one-minute data from the nephelometer and PSAP to invalidate bad data associated with equipment maintenance and malfunction. The edited 1-minute data files were averaged to create the daily averaged data files. In addition, we checked for consistency of data and impossible values such as  $R_{sp}$ ,  $\omega_0 > 1$ . We kept these data in the database but
- omitted them from the plots. Absorption data and the variables depending on absorption (i.e.,  $\omega_0$  and  $\Delta RF/\delta_{ae}$ ) began in 2006. We used AERONET level 2 data, screened for clouds and quality assured by NASA (Holben et al., 2006).

#### **3 Results and Discussion**

First, we describe how we classified the air masses and a few exceptions to the method. Second, we analyse means and variations in climate-forcing properties, discussing how these means and variations support or contradict our hypothesis.

Finally, we discuss the monthly variation of the climate-forcing properties. We focused on a comparison of volcanic ash to other air masses because volcanic ash is a hazard to aviation and our results might serve to detect volcanic ash. We also highlight differences between volcanic ash and African dust because, to the naked eye, these aerosols might look similar, but are different.

# 5 3.1 Aerosol Classification by Source

We present our criteria to classify air masses in Table 1. The method, however, produces uncertainty because we classified average air masses daily but MODIS images are unavailable at night.

# 3.1.1 Clean marine aerosols

Clean marine (CM) aerosols form in the ocean with imperceptible influence from other air masses, natural or anthropogenic, 10 with  $\sigma_{sp} \leq 20 \text{ Mm}^{-1}$ ,  $\sigma_{ap} \leq 0.6 \text{ Mm}^{-1}$ ,  $\delta_{ae} \leq 0.1$ , and trajectories only over the ocean. Trajectories were at 06 and 12Z for 100, 500 and 1000 m above sea level. The clean marine with greater sea salt content (CMS) aerosol met the criteria in CM but,  $\sigma_{sp}$ > 20 Mm<sup>-1</sup> and  $\delta_{ae} > 0.1$ . This definition was based on reports by Kleefeld et al. (2002) on the dependence  $\sigma_{sp}$  with wind speed (square of the speed) and Lewis and Schwartz (2004) that wind speed is a main driver producing natural marine aerosols. VAAC and MODIS images served to verify non-marine aerosols.

# 15 **3.1.2 African dust**

To find African dust (AD), we searched MODIS images for dust clouds leaving western Africa in spring, summer, and fall (see Table 1). If we saw elevated aerosol loads in Puerto Rico about 6-7 days later, we classified the air mass as African dust. An example is the dust cloud over Dakar in May 28, 2010 (Figure 2a), seen over Puerto Rico on June 3, 2010 (Figure 2b). AD in Figures 2a and 2b was, as in other AD events, light brown covering large areas of the Caribbean and Atlantic. AD trajectories were from the east, east-southeast, or southeast. Strong AD episodes such as the event on June 3, 2010 were easily identifiable with MODIS.

## 3.1.3 Volcanic ash

20

We classified an air mass as volcanic ash (VA) if the Soufriere Hills' volcano emitted simultaneously with southeast lowlevel winds, trajectories, and cloud streaks orientation (see Table 1). Figure 3 shows an example during January 9, 2007. We

25 also used cloud streak orientations to estimate the prevailing wind direction because cloud streaks orient parallel to the prevailing winds.

#### 3.1.4 North America and South America air masses

Air masses from North America occurred more in winter or fall, associated with the cold fronts general circulation, as HYSPLIT trajectories suggested. We linked the South America air mass with the broad circulation of cyclones north of Puerto Rico promoting southerly low-level winds. We excluded the North and South America air masses because they only occurred a few times.

occurred a few times.

#### 3.1.5 Exceptions to the classification method

The classification scheme (Table 1) is objective, and a subjective assessment of the classification results led to changes in the assigned classes a few times. As follows, volcanic ash reaches CPR if the Soufriere Hills' volcano emits simultaneously with southeast low-level winds, cloud streaks and trajectories. One exception is when the ash is already in the Atlantic northeast of Puerto Rico (Figure 4). In this instance, winds shifting to the northeast can bring volcanic ash to Puerto Rico.

of Puerto Rico (Figure 4). In this instance, winds shifting to the northeast can bring volcanic ash to Puerto Rico. Also, we used SAL images cautiously because of times that MODIS suggested heavy dust (Figure 5a) over Puerto Rico but SAL images suggested no dust (Figure 5b). Also, Figures 5a and 5b show cloud streaks oriented from northeast to southwest instead of southeast to northwest but we classified the air mass as AD because AD was clear. This example suggests that African dust transport from the Atlantic to CPR.

#### 15 **3.2 Extensive variables by air mass**

#### **3.2.1** Scattering $(\sigma_{sp})$ and absorption $(\sigma_{ap})$ coefficients

Figure 6 shows the  $\sigma_{sp}$  frequency distributions for AD and VA. We verified that these distributions are log-normal by taking the logarithm of the data and applying a normality test. A greater number of values are between ~10 and 100 Mm<sup>-1</sup> with a smaller number of extreme values and empty spaces between them in AD.

Statistically, the empty spaces in the AD frequency distribution suggest that the more extreme AD events might have distinct causes. But studying the potential causes was out of our scope. Alternatively, the extreme events might all have the same cause but the study period was insufficient to fill in the gaps. Note in Figure 6 that in AD,  $\sigma_{sp}$  had a greater range (230 Mm<sup>-1</sup>) in strength compared with that of VA (75 Mm<sup>-1</sup>).

Table 2 summarizes aerosol data collected from 2005-2010. Comparing the aerosol loads among air masses within the MBL, 25 we found that mean  $\sigma_{sp}$  in AD was significantly greater than mean  $\sigma_{sp}$  otherwise (3.5 times greater than CM, 1.9 times greater than CMS with greater sea salt content and 1.7 times greater than VA). Hence, mean atmospheric aerosol load increased, on average, 3.5 times near the surface and the MBL if AD replaced CM. Also, the greater number of AD events ranged from 30 to 35 Mm<sup>-1</sup> and in VA ranged from 15 to 20 Mm<sup>-1</sup>. Mean  $\sigma_{sp}$  in AD and VA differed by 23 Mm<sup>-1</sup> with a 95% confidence interval (CI) from 16.8 to 27.8 Mm<sup>-1</sup>. These results support our hypothesis with respect to aerosol loads.

Concerning columnar data, the volume size distribution for the VA event shown in Figure 7a shows fine aerosols (i.e. not the marine aerosols) dominating the volume (Figure (7b). In contrast, for the VA event shown in Figure 7c, the volume size distribution (Figure 7d) shows coarse aerosols (such as the marine aerosols) dominating the volume. These results imply that VA events are variable enough that sometimes VA dominates the column loading and sometimes sea-salt dominates, implying that two distinct aerosols are present in the VA air mass. However, the optical data in Table 2 shows that VA

5 implying that two distinct aerosols are present in the VA air mass. However, the optical data in Table 2 shows that VA dominates the light scattering during VA events because the light scattering efficiency of coarse-mode particles is much lower than that of fine-mode particles. Therefore, in some VA events, sea-salt might dominate the column mass loading, but VA will dominate the scattering and optical depth.

Turning back to surface data, the ratio of the mean absorption coefficients ( $\sigma_{ap}$ ) between AD and VA suggests that AD absorbs 50% more sunlight than what VA absorbs. To calculate the local heating rate after the absorption by VA or AD we combined the Beer's Lambert law for the absorption rate,  $\frac{dF}{dz} = \frac{\sigma_{ap}}{\mu}F$  with the equation for the local heating rate,  $\frac{dT}{dt} = \frac{1}{\frac{1}{C_p\rho}\frac{dF}{dZ}}$  and obtained,  $\frac{dT}{dt} = \frac{\sigma_{ap}}{\mu C_p\rho}F$ , where *F* is the flux density at altitude z;  $\frac{dF}{dz}$  is the absorption rate;  $\mu$  is the cosine of the zenith angle;  $\sigma_{ap}$  is the absorption coefficient;  $C_p\rho$  is the heat capacity of air (1kJ K<sup>-1</sup> kg<sup>-1</sup>) times  $\rho$ , the density of air; and dT/dt is the local heating rate. The equation for the heating rate is directly proportional to the absorption coefficient. If we substitute the absorption coefficient for AD and VA in the last equation, we get that the change in temperature in AD is 50% greater than the change in temperature of VA. This result only considers the change in temperature due to absorption but omits the effect on temperature due to radiation cooling. An implication of this result is the need to measure the absorption. In the next section, we analyze columnar optical depth, contrasting, or comparing it with the surface scattering.

#### 20 3.2.2 Columnar Aerosol Optical Depth

Similar to the surface mean  $(\sigma_{sp})$  the columnar mean  $(\delta_{ae})$  in AD was greater than that of other air masses. Namely, mean  $\delta_{ae}$  in AD was 5 times greater than that of CM and 3.5 times greater than that of CMS. In contrast,  $\sigma_{sp}$  in AD was 3.5 times greater than that of CM and 1.9 times greater than that of CMS. The different ratios imply a greater AD fraction transported above, than within the MBL. For VA, mean  $\sigma_{sp}$  was 1.1 times greater than that of CMS and the columnar mean  $\delta_{ae}$  in VA

25 was 1.6 times greater than that of CMS. Therefore, VA enhanced the optical depth more than the surface scattering if VA superimposed a CMS air mass. Also, VA enhanced both the scattering and the optical depth twice. Furthermore, mean  $\sigma_{sp}$  in AD was 1.7 times greater than that of VA and mean  $\delta_{ae}$  was 2.1 times greater. Therefore, within the MBL, AD increased the scattering 70% more than that of VA and doubled the columnar optical depth produced by VA. In summary, we found that different air masses affect climate differently because their distinct amounts produce different scattering or extinction.

#### 3.3 Intensive variables by air mass

# 3.3.1 Columnar Aerosol Optical Depth Fine Fraction ( $\delta_{ae}FF$ )

Concerning particle sizes causing extinction in the atmospheric column, an overall mean δ<sub>ae</sub>FF of 0.27 implies that, on average, coarse particles caused three-fourths of the aerosol extinction. In addition, mean δ<sub>ae</sub>FF among air masses was significantly different, except between CM and CMS. This result suggests that in Puerto Rico, different air masses have significantly different columnar size distributions. For instance, the mean δ<sub>ae</sub>FF in VA was 0.09 more than mean δ<sub>ae</sub>FF in AD with a 95% CI from 0.04-0.14. Hence, at p<0.05, columnar VA aerosols were on average, significantly smaller than columnar marine aerosols. In other words, we are more than 95% certain that columnar VA aerosols are smaller than columnar AD or marine aerosols. The mean δ<sub>ae</sub>FF in marine and AD aerosols differed significantly by ~0.03.
0 These results support our hypothesis concerning aerosol sizes. Users of these data should decide if these differences.

10 These results support our hypothesis concerning aerosol sizes. Users of these data should decide if these differences, although significant, are meaningful to them. Our results, in agreement with Reid et al. (2003b), show that coarse aerosols dominate the extinction in Puerto Rico.

#### 3.3.2 Scattering $(R_{sp})$ and Absorption $(R_{ap})$ Fractions

Concerning the sizes of the scattering particles, mean sub-micron scattering fractions, R<sub>sp</sub>, were on average low (Table 2),
with a mean of 0.2. Hence, coarse aerosols produced 80% of the scattering within the MBL. In addition, mean R<sub>sp</sub> in VA was significantly greater than mean R<sub>sp</sub> in AD. Even tough they only differed by 0.03 with a 95% CI from 0.01 to 0.05, the result means that we are more than 95% certain that, within the MBL, volcanic ash has a smaller fraction of coarse scattering particles than what AD and marine aerosols have. The lowest mean R<sub>sp</sub> in marine aerosols implies that marine aerosols have the largest fraction (> 85%) of coarse aerosols. Therefore, the scattering particles sizes among natural aerosols in Puerto Rico
were significantly different, supporting our hypothesis.

Turning to the sizes of the absorbing particles, an overall mean sub-micron absorption fraction,  $R_{ap}$ , of 0.5 shows that on average, coarse and fine absorbing aerosols within the MBL in Puerto Rico have similar amounts. The smaller mean  $R_{ap}$  (0.37) in AD contrasts with the mean  $R_{ap}$ , otherwise, that only differed few hundredths from a mean of 0.6. Hence, coarse absorbing particles in AD produced 63% of the absorption. Therefore, if the absorption by fine AD particles was due

- 25 to soot, or the AD fine mode, or a mix of these two, the result implies that absorption by the coarse iron oxides surpassed the absorption by the fine. A question arising is: Why are absorbing aerosols smaller in VA than in AD if both are iron oxides? One explanation is that VA aerosols are generated by a mechanism in which aerosols are heated to elevated temperature. These results also show that coarse aerosols produced most of the extinction in Puerto Rico within the MBL and support the importance of the SAL as a transport mechanism allowing coarse aerosols to move from the Sahara to the Caribbean and
- 30 farther to Miami (Prospero, 1999), after their lift to high altitudes. Our methods to segregate the sub-micron particles from the total allowed us to measure what fraction of the total scattering and absorption was due to sub-micron aerosols.