# Peer review of "Variations in the physicochemical and optical properties of natural aerosols in Puerto Rico - Implications for climate"

_Atmospheric Chemistry and Physics, 2017_

## Referee Comment (RC1) · Anonymous Referee #1 · 22 Sep 2017

This manuscript uses a combination of observations to classify aerosol air masses over the Cape San Juan Atmospheric Observatory in Puerto Rico. The authors determine the surface and columnar optical properties of the aerosol associated with each of these air masses and do a rough estimate of how these air masses impact the direct radiative forcing.

The overall science in the paper seems sound. I am supportive of this manuscripts publication in ACP once my concerns below have been addressed.

General issues

1) I recommend that the paper be thoroughly proofread for English and general read-

ability. I found a number of sentences that while not technically incorrect were awkward and that took me a while to fully understand. For example, the second sentence of the abstract, "Hundreds of teragrams and the absorbing properties of aerosols such as African dust and volcanic ask affect radiative balance changing atmospheric temperature and thus, climate." took me a long time to parse and fully understand. Further, "hundreds of teragrams" of what? I assume aerosol, but what more specifically? In the global atmosphere? In the column above Puerto Rico? Emitted globally each year? Emitted near Puerto Rico each year? Also, I found a bunch of typos throughout.

2) I'm confused as to why fine-mode aerosol associated with a volcanic aerosol is being called "ash". Ash particles are the primary particles associated with volcanoes and they are overwhelmingly coarse mode (e.g. http://www.nature.com/nature/journal/v528/n7583/fig_tab/nature16153_SF1.html?foxtrotcallback=true, but please search beyond this for "volcanic ash size distribution"). Even accounting for settling before Puerto Rico, these particles should be mostly supermicron. I would guess that most of the fine mode associated with the volcanic plume is actually secondary sulfate associated with the volcano, not ash. Hence, I think it would be better to change the definition of "VA" in the paper to Volcanic Aerosol.

3) What is the relative frequency of each air mass type, and how much does each air-mass type contribute to the radiative forcing over Puerto Rico when accounting for their frequencies?

Specific comments

Abstract: Instruments, such as AERONET, HYSPLIT, and MODIS (sort of), are capitalized as if spelling out the acronym, but then the acronym is not defined in the abstract, which seems strange.

P2 L10: Is the Caribbean different from other regions of the same latitude as acting as a conduit? Is there a citation for this?

[Figure]

P3 L17: Should specify *direct* climate-forcing properties since CCN properties weren't addressed.

P5 L7: "greater the angstrom exponent, the smaller is the size..." This is not strictly true since the angstrom exp oscillates between small positive and negative numbers in the Mie regime.

P6 L6: What are "SAL" images? This is not defined but the "SAL" acronym is used throughout the paper.

P9 L26: "VA enhanced both the scattering the optical depth twice." What does this mean?

Section 3.3.2. I find it somewhat incorrect to refer to the "sizes of absorbing particles" here. It could be that both fine and coarse mode particles are both absorbing, but rather the particles in one mode may be more absorbing that the other. Hence, it would be better to simply saying "there is more absorption from particles in the coarse/fine mode".

P12 L23: Can you explain why the single-scattering albedo doesn't affect the rad forcing efficiency? Is this simply because there isn't enough variability in the single-scattering albedo?

P14 L15: "greater radiative cooling" at the top of the atmosphere or at the surface?
* * *

---

## Referee Comment (RC2) · Anonymous Referee #3 · 16 Nov 2017

General Comments: This paper deals with the variation of the physicochemical and optical properties of natural aerosols over Puerto Rico, as well as with their implication in the climate, in terms of their radiative forcing efficiency. The analysis is based on a database of measurements covering the time period from 2006 to 2010. The authors rightly acknowledges previously reported studies, related to field observations. In general, the manuscript is well written, however some editing in the language is needed. The authors are discussing interesting measurements, and thus it worth being published in Atmospheric Chemistry and Physics. In order to be improved, I would suggest to the authors to take into consideration the following remarks. Specific remarks: 1. Page 1 lines 10-12: I would kindly suggest to the authors to consider moving these

lines from the abstract to the introduction section. 2. In the abstract section, please consider providing also the abbreviations of the corresponding text, where necessary (i.e. Hybrid Single –Particle Lagrangian Integrated Trajectory (Hysplit); Volcanic Ash Advisory Center (VAAC); Moderate Resolution Imaging Spectroradiometer (MODIS); Aerosol Robotic Network (AERONET)). 3. Page 1 lines 28: Please correct to "...more cooling than that of volcanic ash at the top...". 4. The introduction is well written providing a good overview. However, in my opinion in this section should be also mentioned the findings of Saharan dust observations over Caribbean by employing other sensors, like active remote sensing instruments (lidar) (e.g. Burton et al., 2015). 5. Somewhere in the section 2.2, the authors are kindly suggested to state also the name of the AERONET station used in this study, as this is provided in the official AERONET web portal. 6. Page 5 line 22: Correct the typo in the units of the solar constant to "W m-2 ". 7. Page 6 line 1-2: I would kindly suggest to the authors to rephrase this last sentence and give more information regarding the aerosol related products that can be provided by AERONET. For example in this section I would expect to find a statement related to the capability of the sun-sky radiometers to provide the aforementioned aerosol optical properties at various wavelengths, and not only at 500 nm. 8. In general, I would like to suggest to the authors to follow their approach for adjusting the PSAP values to 550 nm, also for the aerosol optical depth measurements obtained by AERONET. In my opinion, for being more accurate in their calculations, the authors are kindly requested to adjust also the AOD values at the wavelength of 550 nm, for decreasing the number of mutual assumptions made for retrieving parameters such as the radiative forcing efficiency. 9. Page 6 line 25: Please rephrase this sentence to something like: "In our dataset, the measurements of absorption along with its related variables, began from 2006". 10. Page 7 line 4: In the last phrase of this sentence is mentioned that African dust and volcanic ash particles are different. This is correct, but the authors have to specify in which sense the aforementioned aerosol types are different, in order to support and strengthen this statement. In any case, my opinion is that meaning of the entire paragraph starting with: "We focused on

.... " up to the this line redundant. The authors have already made clear these issues in the introduction section. 11. Page 7 line 8: The authors would probably like to say "on a 24h-basis" instead of "daily". 12. Page 8 line 11: The authors would probably like to say "because of the existence of specific cases" instead of "of times". 13. Page 8 line 28: The authors would probably like to say "and regarding VA case" instead of "in VA". 14. Page 9 line 1: I have a concern here regarding the characterization of the VA like fine mode aerosols. There are many studies regarding volcanic ash particles that indicates the presence of coarse mode particles inside these plumes. Due to gravitational settling and the meteorological conditions the coarser particles are falling earlier and the finer are traveling longer distances. Thus the age of such aerosols plays a significant role in the size of the finally detected VA particles, and this has to be mentioned in the manuscript. The travelling time of the air mass back trajectories can provide this information of age. 15. Page 9 line 22: Considering correcting "air masses" to "categories". 16. Page 9 line 24: "The different ratios imply a greater AD fraction transported above, than within, MBL". Yes I agree and this statement stands true, however it worth more explanation from the authors. More specifically, it should be stressed more in the manuscript, the height range that the under study aerosol types appear more frequently. This information is passed slightly in the previous paragraphs of Section 3, but only for the marine particles. 17. Page 9 line 28-29: I would suggest that this sentence has to be moved somewhere else in the manuscript (e.g. in the conclusions or in the abstract section). 18. I have a general comment for the section 3.3.4. The authors are kindly requested to discuss more on their finding related to the single-scattering albedo effect on the radiative forcing efficiency. 19. Page 12 line 28-30: These lines are providing unnecessary information. I kindly suggest to the authors to delete the description related to the whisker plot. A reader can easily find this information, if he is not familiar with this kind of plots. 20. Page 14 line 7-8: The first sentence of the conclusion section is misleading. The authors are kindly requested to rephrase it. 21. Page 14 line 29: Replace the word "correct" with the most appropriate word "accurate". 22. Table 1: To my opinion this is not the

appropriate label for this table. The authors are kindly requested to rephrase since this table is not a "Summary of the method to classify . . ." but provides the criteria used for their classification scheme. 23. Following my previous comment I would kindly suggest to the authors to provide a figure demonstrating the flowchart of their classification method. Such a flowchart, in combination with the existing Table 1, will help the reader to go through the manuscript easier. 24. Table 2: The authors are kindly requested to specify here but also in the manuscript, what the standard deviation represents in this table (e.g. systematic? statistical error? daily variability?). Moreover please rephrase as follow: "The left column says" to "The left column indicates"; "at the surface or the columnar" to "and if it refers to surface or atmospheric column". Finally please consider my previous comment for adjusting the AOD values to 550 nm and update the table and the manuscript. 25. I think that the existing Figure 1 is not providing any useful information to the reader, so I would suggest to the authors to delete it. 26. Figure 6: It would be beneficial for the manuscript if the authors present in this figure also significant statistical indicators (e.g. at least mean value and standard deviation). According, to such values that was found in the literature, their classification scheme was based, in order to determine the appropriate threshold values. Moreover, why they demonstrate this frequency distribution only for VA and AD categories and not for the remaining categories (CM & CMS)? 27. Figure 9: The label of y axis (both for Fig. 9 (a)&(b)) is $\Delta$RF/AOD. I suppose that AOD is the aerosol optical depth, however in the text of the manuscript is defined as $\delta$ae. Please correct it, and be consistent through the manuscript.

Please also note the supplement to this comment:
https://www.atmos-chem-phys-discuss.net/acp-2017-703/acp-2017-703-RC2-supplement.pdf